# Evaluation of a novel intervention providing insight into the tobacco industry to prevent the uptake of smoking in school-aged children: a mixed-methods study

Lisa Szatkowski,[1,2] John Taylor,[1,2] Amy Taylor,[1,2] Sarah Lewis,[1,2] Qi Wu,[1,3] Steve Parrott,[1,3] Ann McNeill,[1,4] John Britton,[1,2] Linda Bauld,[1,5] Laura L Jones,[1,6] Manpreet Bains[1,2]

## ABSTRACT

**Objectives** Evidence from the US *Truth* campaign suggests that interventions focusing on tobacco industry practices and ethics may be effective in preventing youth smoking uptake. We developed, piloted and evaluated a school-based intervention based on this premise.

**Methods** Exploratory study students in years 7–8 (aged 11–13) in two UK schools received *Operation Smoke Storm*, comprising three 50 min classroom-based sessions in year 7, an accompanying family booklet and a 1-hour classroom-based booster session in year 8. We compared the risk and odds of ever smoking and susceptibility to smoking in year 8 students in study schools postintervention with students in control schools. Focus groups and interviews with students, teachers and parents evaluated the acceptability of the intervention.

**Results** In intervention schools, the combined prevalence of ever smoking and susceptibility increased from 18.2% in year 7 to 33.8% in year 8. There was no significant difference in the odds of a year 8 student in an intervention school being an ever smoker or susceptible never smoker compared with controls (adjusted OR (aOR) 1.28, 95% CI 0.83 to 1.97, p=0.263) and no significant difference in the odds of ever smoking (aOR 0.82, 95% CI 0.42 to 1.58, p=0.549). Teachers highlighted differences by academic ability in how well the messages presented were understood. Use of the family component was low but was received positively by parents who engaged with it.

**Conclusions** *Operation Smoke Storm* is an acceptable resource for delivering smoking-prevention education, but it does not appear to have reduced smoking and susceptibility.

For numbered affiliations see end of article.

**Correspondence to**
Dr Manpreet Bains;
manpreet.bains@nottingham.ac.uk

### Strengths and limitations of this study

► We used a mixed-methods design that enabled triangulation of quantitative and qualitative data to strengthen the internal and external validity of the findings.
► Conclusions are based on data from only two intervention schools, which served relatively more affluent and ethnically white populations than the national average.
► The comparison with external, non-randomised control data meant that there were significant differences between the characteristics of students in intervention and control schools.
► Logistical difficulties meant we were unable to link students' responses at baseline and follow-up, though smoking behaviours differed little between intervention and control schools at baseline, and analyses were adjusted for confounders measured at follow-up.

## INTRODUCTION

In the UK, nearly 40% of adult smokers start to smoke regularly before the age of 16,[1] and those who start at an early age are more likely to die from a smoking-attributable cause.[2] Therefore, preventing young people from smoking is an important public health priority, and school-based approaches provide an opportunity to reach large numbers of young people. Existing school-based approaches to smoking prevention differ in theoretical approach, design and mode of delivery. However, there is no evidence that any one approach is more superior to another and little conclusive evidence that school-based prevention interventions have anything beyond short-term effects.[3–5] In the only UK study to show significant benefit, training school pupils to initiate conversations about smoking with their peers has been shown to reduce smoking uptake up to 2 years later,[6] though since the publication of this study approaching a decade ago, there have been substantial changes in public attitudes towards smoking as well as in the tobacco control and education environments.

In the USA, the mass media *Truth* campaign has demonstrated some success in encouraging young people not to smoke, focusing on the ethics and exploitative tactics of the tobacco industry.[7–9] Its acceptability and effectiveness has been recognised as worth exploring further in school settings.[5] Previously, we have reported results of a preliminary qualitative evaluation among year 7 students (aged 11–12) in two UK schools of the acceptability of a novel school-based intervention, *Operation Smoke Storm* (*OSS*), based on the premise of *Truth*.[10] Initially, *OSS* comprised three 50 min multimedia interactive teaching sessions, developed by *Kick It*, who delivers the National Health Service Stop Smoking Service for several London boroughs.[11] Further description of this intervention is given in online supplementary file 1.

In focus groups conducted after the delivery of *OSS*, students reported enjoying the intervention and acquiring new knowledge about smoking and the tobacco industry, which seemed to strengthen their aversion to smoking.[10] In one-to-one interviews, teachers expressed confidence delivering the 'off the shelf' resource, although they highlighted a need for the package to be flexible and not dependent on lesson length, teacher confidence or expertise.[10] Following this feedback, year 7 lessons were refined by the research team alongside *Kick It*, primarily to correct technical issues and to increase flexibility and provide teachers with more guidance to help them facilitate discussions regardless of their own level of knowledge. The intervention was also extended to include a family booklet to complement the year 7 lessons to encourage parents to talk to their children about smoking and a 'booster' session for use with year 8 students (aged 12–13) to reinforce the anti-smoking message. These family and booster components are described in online supplementary file 1. Here we report quantitative and qualitative data evaluating the acceptability and effectiveness of the full intervention package.

## METHODS
### Quantitative evaluation
#### Collection of baseline and follow-up data
Six secondary schools in the UK East Midlands region were approached, and two agreed to participate in delivering and evaluating *OSS*. The characteristics of the two schools where *OSS* was delivered are described in detail elsewhere.[10] Personal, Social, Health and Economic Education (PSHE) teachers delivered the first intervention component to all year 7 students in both schools (n=585) in autumn 2013. Before and after intervention delivery, all students were asked to complete an anonymous questionnaire to gather information on their sociodemographic characteristics as well as smoking behaviours and attitudes. Students were asked if they had ever smoked, as well as a set of three previously validated questions to assess their susceptibility to smoking[12]:

1. Do you think that you will try a cigarette soon? (yes/no)

2. If one of your best friends were to offer you a cigarette, would you smoke it? (definitely yes/probably yes/probably not/definitely not)

3. Do you think you will smoke a cigarette at any time during the next year? (definitely yes/probably yes/probably not/definitely not)

Students were classified as non-susceptible if they answered 'no' to the first question 'and 'definitely not' to questions 2 and 3. Students giving other combinations of responses were classified as susceptible.

One year later, in autumn 2014, the booster session was delivered to the same students, then in year 8 (n=538). In school 1, PSHE specialists delivered the booster; 40 min lessons meant they needed two sessions to cover the material. In school 2, changes in the organisation of PSHE meant that the booster was instead delivered by science teachers; lessons here were 1 hour in length, and the material was delivered in a single session. An anonymous questionnaire was administered after the booster session to gather data on smoking behaviours and attitudes and sociodemographic characteristics.

In autumn 2014, the refined year 7 intervention component was also delivered to the new cohort of year 7 students (n=350) in school 1 only, and these students were given the new family booklet to take home. Changes in the delivery of PSHE in school 2 meant that they were not able to accommodate delivery of the year 7 sessions. Questionnaire data were collected at the end of the sessions to gain information about the acceptability of the revised intervention and family component.

### Collection of control data from a non-randomised comparison group
Given some difficulty in recruiting schools and in order to minimise costs, we chose to use external control data collected as part of another study just prior to ours. The Nottingham School Smoking Survey collected data from students in eight schools local to the study area in Spring 2011, 2012 and 2013 (though not all schools participated in every wave). The primary aim of this survey was to evaluate changes in young people's smoking behaviour following the introduction of point-of-sale tobacco display legislation.[13 14] By mid-2013, data were available on current smoking and susceptibility to smoking in year 7 and year 8 for two successive cohorts of students (ie, students who were in year 7 in 2011 and year 8 in 2012 and students who were in year 7 in 2012 and year 8 in 2013).

### Statistical analysis
All data management and analysis was carried out using Stata V.13 (StataCorp). Logistic regression was used to compare the self-reported odds of a combined outcome of ever smoking and susceptibility to smoking, plus ever smoking on its own, in year 8 students after the delivery of the booster session with the odds among year 8 students in the two combined cohorts of students in control schools, using a multilevel model to adjust for clustering

with the effect of school modelled as a random intercept. Due to difficulties in linking students' responses to the year 7 and year 8 questionnaires in intervention schools, ORs could not be adjusted for differences between intervention and control groups at baseline. However, models were adjusted for sociodemographic variables using data collected in year 8, and smoking behaviour at year 7 was compared between intervention and control schools to quantify any differences. Unfortunately, a comparable measure of deprivation was not available across intervention and control schools. Therefore, a proxy indicator of deprivation was created, considering students in the most deprived quintile of the Index of Multiple Deprivation in the control schools and those who reported being eligible for free school meals in the intervention schools as deprived relative to all others. Given the exploratory nature of the study, we have not applied a correction for multiple hypothesis testing but, instead, have presented results with 95% CIs and p values in order to allow the reader to evaluate the findings fully. We also calculated unadjusted and adjusted risk differences (using the 'adjrr' postestimation command in Stata) to compare intervention and control schools.

The non-randomised study was not intended to be fully powered but was instead planned as an exploratory study of the potential effectiveness of the intervention. A prestudy power calculation, based on estimates of the likely achieved sample size in intervention and control schools and the self-reported prevalence of ever smoking and susceptibility among year 8 students, suggested that we would be able to estimate the risk difference to within 6.6%, that is, if the observed effect was 6.7% or greater the CIs would preclude the possibility of no effect or a negative effect of the intervention. This effect size was consistent with the size of effect that a subsequent cluster-randomised controlled trial would be powered to detect and in line with the size of effect used to power the A stop smoking in schools trial study (ASSIST).[6] For each of our outcomes (ever smoking and susceptibility to smoking, plus ever smoking on its own), we also calculated Bayes factors under three different scenarios in order to assess whether our data provided substantial evidence for or against the null hypothesis: (1) assuming a maximum OR of 2, that is, a doubling of never smokers in intervention compared with control schools, taking hypothesised values uniformly distributed between 0 and the maximum as plausible values; (2) assuming a plausible predicted OR of 2 and taking hypothesised values in a normal distribution around this value; and (3) assuming a plausible predicted OR of 2 and taking hypothesised values in a half normal distribution around this value. A Bayes factor of 3 or more was taken as substantial evidence against the null hypothesis and 1/3 or less as evidence for the null.

We have followed the STrengthening the Reporting of OBservational studies in Epidemiology statement in reporting the results of this study.

## Qualitative evaluation
### Focus group and interview procedures
The qualitative evaluation comprised focus groups with year 7 and year 8 students, interviews with teachers who delivered the year 7 sessions and the year 8 booster session and paired year 7 student–parent interviews to evaluate the family booklet, each guided by a semistructured interview schedule. We used the same procedures as described previously.[10] In summary, we conducted two gender-specific focus groups with year 7 students in the one school (school 1) that delivered the revised sessions (16 students in total: 8 male, 8 female) and 8 focus groups with year 8 students across the two schools (51 students in total: 25 male, 26 female). Students shared their views on the sessions and their awareness of and attitudes towards the tobacco industry and smoking. Both year 7 focus groups lasted for 26 min and year 8 focus groups lasted for 24 min on average (range 11–35 min). All year 7 and year 8 teachers who delivered part of the intervention were invited by email to be interviewed about its acceptability and effectiveness; 10 year 7 teachers and 6 year 8 teachers took part (4 from school 1, 2 from school 2; interviews lasted 26 min on average (range 19–33 min)). The family booklet was accompanied by a letter inviting parents to express an interest in participating in a paired student–parent interview to explore their views. These interviews sought students' and parents' views on the booklet and how they engaged with it. An inconvenience allowance (£15 high-street voucher) was offered to each pair who participated (n=9). Interviews took place in participants' home or on school premises according to individual preference (lasted 23 min on average and ranged between 13 and 33 min).

### Data analysis
Analysis procedures were similar to those used previously,[10] which followed the framework approach.[15 16] Digital audio-recordings were transcribed verbatim. A sample of focus group and interview transcripts was read initially (by AT and JT) to identify initial codes, themes and subthemes and any within-group or between-group differences (school and gender). As in our earlier work, codes identified from the focus groups, teacher interviews and student–parent interviews were similar (apart from teachers' interview data identifying a theme about preparation to deliver the intervention), and thus, all year 7 data were analysed together and similarly all year 8 data. Initial themes and subthemes were discussed between the researchers (AT, JT, MB, LS) to reach consensus on an initial analytical framework. This framework was applied and refined following analysis of the remaining transcripts and until the point of data saturation. Data were then indexed according to the final framework and charted into matrices according to each theme to facilitate synthesis and interpretation.

Similar themes were identified for both the year 7 and year 8 intervention components, and these supported those reported in our initial evaluation[10]: *Teachers'*

*preparedness to deliver OSS; Raised awareness; Engagement with the intervention*; and *Options for extending the resource* (see online supplementary file 2 for details of themes). Qualitative findings with respect to the year 7 sessions were similar to those reported previously,[10] and the amendments made to correct technical issues, increase flexibility and provide teachers with more guidance were positively received. Therefore, the qualitative findings presented here focus on evaluation of the family booklet and year 8 booster session.

### Ethics and consent

Parents of students in both year 7 and year 8 were sent a letter informing them about *OSS* and the accompanying academic evaluation, approximately 3 weeks prior to delivery. They were asked to return an opt-out slip if they did not want their child to complete a questionnaire or to participate in a focus group. Students were able to opt out of questionnaire completion and were under no obligation to volunteer for focus groups. Written informed consent was obtained from participants prior to data collection.

## RESULTS

### Did *OSS* have an impact on smoking behaviour?

Completed questionnaires were received from 445 year 8 students in intervention schools and 1692 year 8 students in control schools; table 1 describes students' characteristics.

As expected, given the non-randomised nature of the study, there were significant differences between students in intervention and control schools. In control schools, a greater proportion of students were of non-white ethnicity, had parents who smoked, reported smoking was allowed in their home, had more friends who smoked and were ever smokers themselves.

Table 2 shows the odds of a student being a susceptible never smoker and/or an ever smoker in year 7 and year 8 in the two intervention schools compared with control schools. After adjusting for significant confounders, there were no differences in ever smoking and susceptibility to smoking between intervention and control schools in year 7. In year 8, after adjusting for significant confounders, the odds of a student in an intervention school being an ever smoker or susceptible never smoker were 28% higher than the odds for a student in a control school, though this difference was not statistically significant (adjusted OR 1.28, 95% CI 0.83 to 1.97, p=0.263). The adjusted risk difference suggested a non-significant 4.1% higher prevalence of ever smoking and susceptibility to smoking in intervention schools. Students in intervention schools were slightly less likely to have ever smoked compared with students in control schools, though again the difference was not statistically significant (adjusted OR 0.82, 95% CI 0.42 to 1.58, p=0.549). The adjusted risk difference suggested a non-significant 2.0% lower prevalence of ever smoking OSS in intervention schools.

**Table 1** Characteristics of year 8 students in intervention and control schools

| | Intervention schools, n (%) | Control schools, n (%) | p Value for difference* |
|---|---|---|---|
| Total number of completed questionnaires received | 445 | 1692 | |
| **Sex** | | | |
| Male | 200 (44.9) | 843 (49.8) | 0.482 |
| Female | 216 (48.5) | 843 (49.8) | |
| Missing | 29 (6.5) | 6 (0.4) | |
| **Ethnic group** | | | |
| White | 368 (82.7) | 1309 (77.4) | <0.001 |
| Non-white | 27 (6.1) | 220 (13.0) | |
| Missing | 50 (11.2) | 163 (9.6) | |
| **Parental smoking** | | | |
| Neither | 302 (67.9) | 1123 (66.4) | 0.031 |
| At least one | 106 (23.8) | 516 (30.5) | |
| Missing | 37 (8.3) | 53 (3.1) | |
| **Sibling smoking** | | | |
| None | 365 (82.0) | 1461 (86.4) | 0.852 |
| At least one | 43 (9.7) | 178 (10.5) | |
| Missing | 37 (8.3) | 53 (3.1) | |
| **Smoking in the home** | | | |
| Not allowed | 369 (82.9) | 1460 (80.4) | <0.001 |
| Allowed | 36 (7.6) | 375 (16.3) | |
| Missing | 42 (9.4) | 57 (3.4) | |
| **Number of friends who smoke** | | | |
| None | 289 (64.9) | 734 (43.4) | <0.001 |
| One or two | 48 (10.8) | 236 (14.0) | |
| Three or more | 18 (4.0) | 254 (15.0) | |
| Missing | 90 (20.2) | 468 (27.7) | |
| **Rebelliousness and sensation seeking[17]** | | | |
| Low | 225 (50.6) | 870 (51.4) | 0.661 |
| High | 176 (39.6) | 715 (42.3) | |
| Missing | 44 (9.9) | 107 (6.3) | |
| **Academic performance (self-perceived)** | | | |
| Excellent or good | 313 (70.3) | 1228 (72.6) | 0.372 |
| Average or below average | 92 (20.7) | 406 (24.0) | |
| Missing | 40 (9.0) | 58 (3.4) | |
| **Eligible for free school meals** | | | |
| No | 374 (84.0) | Not collected | N/A |
| Yes | 25 (5.6) | | |
| Missing | 46 (10.3) | | |
| **Index of Multiple Deprivation quintile** | | | |

Continued

**Table 1** Continued

| | Intervention schools, n (%) | Control schools, n (%) | p Value for difference* |
|---|---|---|---|
| Least deprived | Not collected | 375 (22.2) | N/A |
| 2 | | 160 (9.5) | |
| 3 | | 282 (16.7) | |
| 4 | | 240 (14.2) | |
| Most deprived | | 261 (15.4) | |
| Missing | | 374 (22.1) | |

*Excluding missing data.

Bayes factors for the combined outcome were 1/3 or lower under each of the three scenarios tested, suggesting that our data provide substantial evidence for the null hypothesis of no positive effect of the intervention. Bayes factors for ever smoking were all close to 1, suggesting that our data are insensitive and unable to distinguish between the alternative and null hypotheses.

### What did students, teachers and parents think about *OSS*?

Students broadly liked *OSS*; 77.1% of year 7 students said that the revised year 7 sessions were very good or okay and 72.4% of year 8 students evaluated the booster session similarly. Qualitative data from year 8 focus groups showed that the booster session was well received and that most students bought into the storyline (box 1 points a and b).

Of the 61.6% of year 7 students who reported receiving the family booklet and taking it home, 43.0% said they showed it to their mother or another adult female, 21.5% reported showing to their father or another adult male and 24.4% said that they did not show the booklet to anyone. Very few reported having completed activities with a parent or carer. Even though year 7 students and parents who were interviewed endorsed the family booklet as a way to improve knowledge and initiate conversations around smoking (box 1 points c and d), our qualitative data also indicated that often the booklet was not used as intended—many students simply did not show the booklet to their parents or realise the booklet was for them to complete with their parents (box 1 points e and f).

### Did *OSS* change students' knowledge and attitudes about smoking?

69.3% of year 7 students and 45.0% of year 8 students thought that *OSS* had made it less likely that they would ever try a cigarette. Students displayed some changes in knowledge and attitudes over the course of the study (table 3).

Qualitative findings from year 8 students and teachers suggest that the booster session raised awareness of the harmful effects of tobacco and some students showed an appreciation of why and how the tobacco industry might target young people (box 2 points a and b). However, teachers mentioned that not all students understood

**Table 2** ORs and adjusted risk differences for smoking outcomes in intervention compared with control schools

| | Prevalence (%) | | Unadjusted | | | Adjusted* | | |
|---|---|---|---|---|---|---|---|---|
| | Students in intervention schools | Students in control schools | Odds of outcome in intervention vs control schools OR (95% CI) | p Value | Risk difference % (95% CI) | Odds of outcome in intervention vs control schools OR (95% CI) | p Value | Risk difference % (95% CI) |
| **Year 7 (before intervention delivery)** | | | | | | | | |
| Ever smoker or susceptible never smoker | 18.2 (92/505) | 22.9 (351/1530) | 0.82 (0.43 to 1.55) | 0.536 | −4.7 (−15.3 to 5.9) | 1.74 (0.54 to 5.56) | 0.351 | 5.9 (−13.8 to 2.6) |
| Ever smoker | 2.4 (12/505) | 6.4 (98/1530) | 0.38 (0.13 to 1.08) | 0.070 | −4.0 (−6.9 to 1.2) | 1.22 (0.13 to 11.3) | 0.858 | 0.4 (−9.9 to 10.8) |
| **Year 8 (after intervention delivery)** | | | | | | | | |
| Ever smoker or susceptible never smoker | 33.8 (145/429) | 30.9 (504/1631) | 1.17 (0.70 to 1.95) | 0.556 | 2.9 (−4.0 to 9.8) | 1.28 (0.83 to 1.97) | 0.263 | 4.1 (−0.5 to 8.6) |
| Ever smoker | 7.9 (34/429) | 10.7 (175/1631) | 0.80 (0.32 to 1.98) | 0.622 | −2.8 (−7.8 to 2.1) | 0.82 (0.42 to 1.58) | 0.549 | −2.0 (−5.4 to 1.4) |

*Adjusted for: perceived academic ability; rebelliousness; sibling smoking; parental smoking; and whether smoking is allowed in the family home.

| Box 1 What did students, teachers and parents think about *OSS*? |
| --- |
| a. *It was really good. It was something different and I liked it.* (School 2, F) |
| b. *They're [the videos] really cool because I like when that girl went on a mission. That was, kind of like, interesting because I was like, 'What is she going to do next?'* (School 1, F) |
| c. *I learned something, I didn't know about all the additives if you like; and the sneaky way that the big companies and the amount of money involved and all of that really.* (School 1, Parent 1) |
| d. *We've discussed it since and had a chat about it. We were talking about it the other day, weren't we, things like the booklet and things like that and talking about what we now know about it. It was building on really what you'd done in [Drug Abuse Resistance Education] DARE at primary, wasn't it, just taking it a bit further.* (School 1, parent 7) |
| e. *My tutor didn't really explain what it actually was about, so I didn't know I actually had to do anything with it, that's why I didn't show my mum.* (School 1, parent 3) |
| f. *That's why I just thought, 'Oh, it's for my parents, it's not for me.'* (School 1, F) |

| Box 2 Did *OSS* change students' knowledge and attitudes about smoking? |
| --- |
| a. *I didn't know about like all the effects until this year, and it's just like, it just shows you what actually smoking does. It just opened my eyes a bit.* (School 1, F) |
| b. *If they target to young people and try and get to young people, then they will get more money, 'cause there'll be more people getting addicted to it.* (School 2, F) |
| c. *Do you know that little clip where the boss is being very subtle going, 'Oh do you use social media?' And, 'Oh we could do brand placement. Oh but we're not allowed to.' And it was all very subtle… Yeah and do you know lower-ability pupils wouldn't have got that. I think that would have confused them, where the other pupils it wouldn't have.* (School 1, teacher 1) |
| d. *So whenever they talked about like their Tweets for social media, they kind of went for, 'You shouldn't smoke, it's bad for you. You shouldn't smoke. Cigarettes have got all this stuff in them,' so kind of the obvious stuff from it, but they then don't take it that step further to think, like, should they be publicising it, yeah, taking that conversation a bit deeper.* (School 1, teacher 2) |

this message and highlighted differences in the extent to which students of higher and lower academic abilities could remember the new information and complete the activities (box 2 points c and d).

## DISCUSSION

This project was the first to formally evaluate a school-based smoking prevention intervention highlighting the ethics and exploitative tactics of the tobacco industry. The intervention was feasible to deliver in the classroom, was generally acceptable to teachers, students and parents and helped to raise awareness about smoking-related issues and the tobacco industry. However, there was no significant difference in the odds or risk of self-reported ever smoking and susceptibility to smoking in students who received *OSS* compared with students from local schools where the intervention was not delivered.

Synthesis of the quantitative and qualitative data offers potential suggestions as to why the intervention does not appear to be effective in preventing smoking uptake. In both the focus groups with year 7 students reported previously[10] and those following delivery of the revised year 7 sessions, students' interest and recall centred mainly on the chemical constituents of cigarettes and/or the health effects of smoking. There was some suggestion from teachers that concepts relating to tobacco marketing, particularly where they were mentioned more subtly, were too advanced for students of lower academic ability to fully grasp. Given that educational attainment is inversely associated with adolescent smoking,[17] it might be that *OSS* failed to reach those students most likely to become smokers.

The prevalence of smoking among young people increases with age,[18] and it might be that any effect of *OSS* on uptake is delayed beyond the follow-up period studied

| Table 3 Mean Likert scale responses (1=strongly agree, 5=strongly disagree) | | | | |
| --- | --- | --- | --- | --- |
| | How far do you agree with the following statements? (mean+SD for statements 1–3; median+IQR for statement 4) | | | |
| | Baseline | After year 7 lessons in phase 1 | After year 8 lessons in phase 2 | p Value* |
| 1. Companies that make cigarettes only try to attract customers aged 18+ | 2.30 (1.04) | 2.85 (1.22) | 3.47 (1.07) | <0.001 |
| 2. Companies that make cigarettes sell dangerous products but still operate in a fair and decent way | 2.79 (0.95) | 2.80 (1.04) | 2.95 (0.95) | 0.030 |
| 3. Smoking is not that serious compared with other drugs young people use | 3.06 (1.13) | 3.20 (1.16) | 3.24 (1.09) | 0.034 |
| 4. Nicotine in cigarettes is one of the most addictive drugs that people use | 2 (1–3) | 2 (1–2) | 2 (1–3) | <0.001 |

*ANOVA F test for normally distributed variables, Kruskal-Wallis test for non-normally distributed variables.
ANOVA F, analysis of variance.

here. Many students reported that participation in *OSS* had made it less likely that they would try a cigarette, and there was evidence of increasing disagreement over time with statements such as 'Smoking is not that serious compared with other drugs that young people use'. These data are encouraging, and although these shifts in attitudes are not reflected in self-reported smoking and susceptibility in year 8, the possibility remains that the impact of the intervention may become evident among these students in years to come.

The year 8 students on whom the primary analysis is based received the original version of the year 7 lessons that were subsequently revised. Therefore, it is possible that the effect of the revised resources on smoking and/or susceptibility might have been different. However, given the fact that the majority of the changes made were to correct technical issues rather than changes to content, this is unlikely. In addition, year 8 students had not received the family component of the intervention. However, few year 7 students in the second phase of the study used the booklet as intended so it is unlikely that this would have a substantial effect on the outcomes.

The 95% CIs around the ORs quantifying differences in smoking behaviours between students in intervention and control schools were wide, and the adjusted risk differences were small. The direction of the point estimate for the odds of ever smoking tentatively suggests that exposure to *OSS* might reduce the odds of this outcome, although the OR for the combined outcome of ever smoking plus susceptibility suggests an increase in odds. A reduction in ever smoking following exposure to *OSS* would be encouraging, and with a larger sample size, the precision of the effect estimates would improve and smaller effect sizes may be detected as statistically significant.

The study findings are based on data from only two schools and may not be generalisable to schools more widely, particularly with regard to students' ethnicity and deprivation. The non-randomised comparison meant that there were significant differences between the characteristics of students in intervention and control schools, which we were not able to adjust for. Our conclusions also rely on self-reported data, even though measures such as ensuring students' anonymity were in place to encourage honest responses.

The use of topic guides and the rigorous analytical process of the framework approach counterbalanced any potential for biased interpretation in favour of the intervention. However, some year 8 focus groups had a small number of participants, which meant there was a less-than-ideal group dynamic. Finally, the students, teachers and parents who took part in the focus groups and interviews were a self-selecting sample, which introduced potential for bias.

Despite there being no evidence of effectiveness in this study, there is scope for further work to understand whether the concept behind *OSS* is worth pursuing further. *OSS* as it stands is probably not suitable for use with students older than the year 7 and year 8 groups, but the concept might be effective if used as the basis of an age-appropriate intervention with older students who might be better able to engage with subtle messages about industry influences. Alternatively, *OSS* might usefully be adapted to include fully differentiated activities and resources for use with different academic abilities. Given the erosion of PSHE within the curriculum, there is scope to understand whether *OSS* could be delivered effectively in other settings such as youth groups. Finally, further work is warranted to explore how to effectively engage parents and guardians more in supporting their child to remain smoke free.

**Author affiliations**
[1]UK Centre for Tobacco and Alcohol Studies, University of Nottingham, Nottingham, UK
[2]Division of Epidemiology and Public Health, Nottingham City Hospital, University of Nottingham, Nottingham, UK
[3]Department of Health Sciences, University of York, York, UK
[4]National Addictions Centre, King's College London, Institute of Psychiatry, Psychology and Neuroscience, London, UK
[5]Institute for Social Marketing, University of Stirling, Stirling, UK
[6]Department of Public Health, Epidemiology & Biostatistics, School of Health and Population Sciences, College of Medical and Dental Sciences, University of Birmingham, Birmingham, UK

**Acknowledgements** We thank the personal, social, health and economic education teaching leads, teachers and students at the schools who took part in this study, as well as the chair and members of the study steering committee and data monitoring and evaluation committee for their valuable advice and support. We also thank Tomasz Letniowski and Toby Fairs-Billam at Kick It who developed *Operation Smoke Storm* but who were independent of this evaluation.

**Collaborators** UK Centre for Tobacco and Alcohol Studies.

**Contributors** LS, MB, SL, AM, JB, LLJ, LB, SP and QW designed the study. JT and AT led the focus groups and conducted the teacher interviews; MB, LS and LLJ observed the focus groups. JT and AT analysed the data with MB providing external validation of themes. LS and JT wrote the first draft of this manuscript. All authors made critical comments on subsequent drafts of the paper and have approved the final version.

**Funding** This project was funded by the National Institute for Health Research Public Health Research (NIHR PHR) Programme (grant number 11/3010/02). The views and opinions expressed herein are those of the authors and do not necessarily reflect those of the NIHR PHR Programme or the Department of Health. The authors are members of the UK Centre for Tobacco and Alcohol Studies, a Public Health Research Centre of Excellence funded by the UK Clinical Research Collaboration. Funding from the British Heart Foundation, Cancer Research UK, the Economic and Social Research Council, the Medical Research Council and the National Institute of Health Research, under the auspices of the UK Clinical Research Collaboration, is gratefully acknowledged.

**Competing interests** LLJ receives personal fees from the National Centre for Smoking Cessation and Training, outside the submitted work.

**Ethics approval** University of Nottingham Medical School Research Ethics Committee (reference 13122012 CHS EPH Smoking).

**Provenance and peer review** Not commissioned; externally peer reviewed.

**Data sharing statement** Requests for access to data should be addressed to the corresponding author.

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
