## [Reviewer comments · BMJ Open]

ARTICLE DETAILS

TITLE (PROVISIONAL)	Evaluation of a novel intervention providing insight into the tobacco industry to prevent the uptake of smoking in school-aged children: a mixed-methods study
AUTHORS	Szatkowski, Lisa; Taylor, John; Fuller, Amy; Lewis, Sarah; Wu, Qi; Parrott, Steve; McNeill, Ann; Britton, John; Bauld, Linda; Jones, Laura; Bains, Manpreet

VERSION 1 – REVIEW

REVIEWER	Lilian Krist Charité-Universitätsmedizin Berlin Institute for Social Medicine, Epidemiology and Health Economics Berlin, Germany No Competing Interest
REVIEW RETURNED	12-Jun-2017

GENERAL COMMENTS	The authors of this manuscript use a mixed-methods approach to address an interesting topic. Unfortunately they decided to take an existing sample of student data as control group instead of recruiting an appropriate control group themselves. However, they address this issue in the discussion part. Introduction The introduction part is very short. I would like to know some more research results. The authors describe in detail their previous study, it would be interesting to have a broader image of the actual state of research concerning smoking prevention studies. Introduction and methods are somewhat mixed up. Intervention is partly described in the introduction part. Introduction page 5 lines 50-57 and page 6 lines 3-25 are methods. Methods Personal, Social, Health and Economic Education (PSHE) is not explained at the beginning. Abbreviations have to be explained the first time they are used. Recruitment process should be briefly described, even if the authors refer to the detailed description in another paper. Description of the questions the students had to answer is a little confuse. Better: list all three questions one after another with the respective answer possibilities. Was the school where the new cohort of 7 year students were provided with the refined intervention one of the schools that participated in the first „wave“? Or is it a new school? This should be better explained. (Later the authors write „school 1“ and „school 2“ – this should be done earlier in the manuscript.)
---

REVIEWER	Roger Thomas Department of Family Medicine Faculty of Medicine University of Calgary, Alberta, Canada.
REVIEW RETURNED	29-Jun-2017

GENERAL COMMENTS	This is an interesting and meritorious project. You correctly outlined the limitations (due to resources) of your project. It would be inappropriate to draw conclusions about the effectiveness of your intervention from a non-randomised design (with clear differences in factors related to the outcome measures in the groups), unknown confounders, the need to use data from another study (Nottingham School Smoking Survey), and develop a proxy measure of deprivation. I would drastically shorten the description of the evaluation of the intervention but keep the key statements about how you evaluated the project. The field needs new student-oriented interventions. I was really interested to learn from this (large) pilot about the content of your intervention and the students', teachers' and parents' evaluation and responses. However, I learned nothing about the content of your intervention, how you developed it and tested it, elements you discarded or strengthened, completeness of delivery to students as measured by adherence to a manual, and what you learned from observations in the classroom about how the students responded and what issues they picked up on and responded to. I do not know how I could improve and test your intervention with other populations. We learn only that "the students broadly liked OSS," that they shared it more with mothers/other females than fathers/other males but few activities with parents occurred. Please share more about your interesting project.
---

REVIEWER	Roger Thomas Department of Family Medicine Faculty of Medicine University of Calgary, Alberta, Canada.
REVIEW RETURNED	29-Jun-2017

GENERAL COMMENTS	This is an interesting and meritorious project. You correctly outlined the limitations (due to resources) of your project. It would be inappropriate to draw conclusions about the effectiveness of your intervention from a non-randomised design (with clear differences in factors related to the outcome measures in the groups), unknown confounders, the need to use data from another study (Nottingham School Smoking Survey), and develop a proxy measure of deprivation. I would drastically shorten the description of the evaluation of the intervention but keep the key statements about how you evaluated the project. The field needs new student-oriented interventions. I was really interested to learn from this (large) pilot about the content of your intervention and the students', teachers' and parents' evaluation and responses. However, I learned nothing about the content of your intervention, how you developed it and tested it, elements you discarded or strengthened, completeness of delivery to students as measured by adherence to a manual, and what you learned from observations in the classroom about how the students responded and what issues they picked up on and responded to. I do not know how I could improve and test your intervention with other populations. We learn only that "the students broadly liked OSS," that they shared it more with mothers/other females than fathers/other males but few activities with parents occurred. Please share more about your interesting project.
---

VERSION 1 – AUTHOR RESPONSE

Reviewer: 1

Comment: The introduction part is very short. I would like to know some more research results. The authors describe in detail their previous study, it would be interesting to have a broader image of the actual state of research concerning smoking prevention studies.

Response: Whilst we agree that it would be interesting to present a broader view of the current state of research, we are close to the journal word limit for the article and are therefore restricted in the level of detail we can present. We already cite the key systematic reviews of the field, including a Cochrane review (ref 3) and a summary of evidence for the UK National Institute for Health and Clinical Evidence (ref 5). We have added an additional statement to the effect that existing approaches are highly heterogeneous and there is no evidence that any one approach is superior. In addition, we also now mention the only UK study to have shown a significant reduction in smoking uptake (ref 6). This section now reads:

“Existing school-based approaches to smoking prevention differ in theoretical approach, design and mode of delivery. However, there is no evidence that any one approach is more superior to another, and little conclusive evidence that school-based prevention interventions have anything beyond short-term effects 3-5. In the only UK study to show significant benefit, training school pupils to initiate conversations about smoking with their peers has been shown to reduce smoking uptake up to two years later⁶, though since the publication of this study approaching a decade ago there have been substantial changes in public attitudes towards smoking as well as in the tobacco control and education environments.”

Comment: Introduction and methods are somewhat mixed up. Intervention is partly described in the introduction part. Introduction page 5 lines 50-57 and page 6 lines 3-25 are methods.

Response: The sections of text on p5 lines 50-57 and p6 lines 3-25 relate to the findings of the previous study and have been used to set the context for the current work. As these sections do not relate specifically to the methods of the current study we therefore feel that they sit better within the introduction.

Comment: Personal, Social, Health and Economic Education (PSHE) is not explained at the beginning. Abbreviations have to be explained the first time they are used.

Response: We have now defined this acronym at its first use (p7)

Comment: Recruitment process should be briefly described, even if the authors refer to the detailed description in another paper.

Response: We have amended this sentence as follows (p7):

“Six secondary schools in the UK East Midlands region were approached and two agreed to participate in delivering and evaluating OSS. The characteristics of the two schools where OSS was delivered are described in detail elsewhere”

Comment: Description of the questions the students had to answer is a little confuse. Better: list all three questions one after another with the respective answer possibilities.

Response: We have amended the wording of this section as follows (p7):

“Students were asked if they had ever smoked, as well as a set of three previously-validated questions to assess their susceptibility to smoking:

1) do you think that you will try a cigarette soon? (yes/no)

2) if one of your best friends were to offer you a cigarette, would you smoke it? (definitely yes/ probably yes/ probably not/ definitely not)

3) do you think you will smoke a cigarette at any time during the next year? (definitely yes/ probably yes/ probably not/ definitely not)

Students were classified as non-susceptible if they answered ‘no’ to the first question ‘and ‘definitely not’ to questions two and three. Students giving other combinations of responses were classified as susceptible.”

Comment: Was the school where the new cohort of 7 year students were provided with the refined intervention one of the schools that participated in the first „wave“? Or is it a new school? This should be better explained. (Later the authors write „school 1“ and „school 2“ – this should be done earlier in the manuscript.)

Response: Yes, this took place in the same school that participated in the first wave. We have now used the terms school 1 and school 2 throughout in order to remove any confusion.

Reviewer: 2

Comment: It would be inappropriate to draw conclusions about the effectiveness of your intervention from a non-randomised design (with clear differences in factors related to the outcome measures in the groups), unknown confounders, the need to use data from another study (Nottingham School Smoking Survey), and develop a proxy measure of deprivation. I would drastically shorten the description of the evaluation of the intervention but keep the key statements about how you evaluated the project.

Response: We are not clear to which part of the manuscript this comment is referring. We have been as succinct as possible in describing how we used quantitative and qualitative methods to evaluate the intervention, and in the discussion section we evaluate the strengths and weaknesses of the project itself in some depth.

Comment: The field needs new student-oriented interventions. I was really interested to learn from this (large) pilot about the content of your intervention and the students', teachers' and parents' evaluation and responses. However, I learned nothing about the content of your intervention, how you developed it and tested it, elements you discarded or strengthened, completeness of delivery to students as measured by adherence to a manual, and what you learned from observations in the classroom about how the students responded and what issues they picked up on and responded to.

Response: With limits on word count we were unable to describe the content of the intervention and the process of its development in any great depth within the body of the article itself. However, the components are described in detail in Supplementary file 1 which is referred to at the end of the introductory section. There was no ‘manual’ as such to measure completeness of delivery; this issue was addressed within the teacher interviews where there was nothing to suggest the intervention was generally delivered as intended.

We were unable to carry out direct observation of intervention delivery in the classroom (and indeed doing so may have led to a Hawthorne effect) but the student focus groups and teacher interviews gave insight into the students' responses and the issues that were picked up.

Comment: I do not know how I could improve and test your intervention with other populations.

Response: We have briefly discussed potential improvements to the intervention in our Discussion section (p18), such as the inclusion of more extensively differentiated activities, and mention the merit of further work to understand whether the intervention may be delivered in settings other than schools, such as youth groups. We do not feel that we have the scope, within the allowable word limit, to go into great detail about future work.

Comment: We learn only that "the students broadly liked OSS," that they shared it more with mothers/other females than fathers/other males but few activities with parents occurred. Please share more about your interesting project.

Response: Again, we were limited within the word count in the amount of detail we have been able to give. We have provided quotes from our qualitative data that succinctly summarise the key findings and triangulate with the quantitative data.